# The Hepatokine RBP4 Links Metabolic Diseases to Articular Inflammation

**DOI:** 10.3390/antiox13010124

**Published:** 2024-01-19

**Authors:** Andrés Pazos-Pérez, María Piñeiro-Ramil, Eloi Franco-Trepat, Ana Alonso-Pérez, María Guillán-Fresco, Antía Crespo-Golmar, Miriam López-Fagúndez, Javier Conde Aranda, Susana Belen Bravo, Alberto Jorge-Mora, Rodolfo Gómez

**Affiliations:** 1Musculoskeletal Pathology Group, Health Research Institute of Santiago de Compostela (IDIS), Santiago University Clinical Hospital, SERGAS, 15706 Santiago de Compostela, Spain; andres.pazos.perez@sergas.es (A.P.-P.); maria.pramil@udc.es (M.P.-R.); eloi.franco.trepat@sergas.es (E.F.-T.); ana.alonso.perez@sergas.es (A.A.-P.); maria.guillan.fresco@sergas.es (M.G.-F.); antia.crespo.golmar@sergas.es (A.C.-G.); miriam.lopez.fagundez@sergas.es (M.L.-F.); susana.belen.bravo@sergas.es (S.B.B.); alberto.agustin.jorge.mora@sergas.es (A.J.-M.); 2Molecular and Cellular Gastroenterology, Health Research Institute of Santiago de Compostela (IDIS), Santiago University Clinical Hospital, SERGAS, 15706 Santiago de Compostela, Spain; javier.conde.aranda@sergas.es

**Keywords:** arthritis, chondrocytes, gout, inflammation, crystal arthropathies

## Abstract

Objectives: This study investigates the role of retinol binding protein 4 (RBP4) in an articular context. RBP4, a vitamin A transporter, is linked to various metabolic diseases. Methods: Synovial fluid RBP4 levels were assessed in crystalline arthritis (CA) patients using ELISA. RBP4’s impact on articular cell types was analysed in vitro through RT-PCR and flow cytometry. Proteomic analysis was conducted on primary human osteoarthritis chondrocytes (hOACs). Results: Synovial fluid RBP4 concentrations in CA patients correlated positively with glucose levels and negatively with synovial leukocyte count and were elevated in hypertensive patients. In vitro, these RBP4 concentrations activated neutrophils, induced the expression of inflammatory factors in hOACs as well as synoviocytes, and triggered proteomic changes consistent with inflammation. Moreover, they increased catabolism and decreased anabolism, mitochondrial dysfunction, and glycolysis promotion. Both in silico and in vitro experiments suggested that RBP4 acts through TLR4. Conclusions: This study identifies relevant RBP4 concentrations in CA patients’ synovial fluids, linking them to hypertensive patients with a metabolic disruption. Evidence is provided that RBP4 acts as a DAMP at these concentrations, inducing robust inflammatory, catabolic, chemotactic, and metabolic responses in chondrocytes, synoviocytes, and neutrophils. These effects may explain RBP4-related metabolic diseases’ contribution to joint destruction in various rheumatic conditions like CA.

## 1. Introduction

Retinol binding protein 4 (RBP4) is a hepatokine member of the lipocalin family which is characterized by its carrier function of small hydrophobic molecules in the blood. It was first described as a plasma protein that binds retinol (vitamin A) and transports it through the bloodstream [1]. When we refer to RBP4, we must make a differentiation between the two isoforms of holo-Rbp4, when it is bound to retinol, and apo-Rbp4, when it is found alone. In its holo form, Rbp4 binds to another molecule called transthyretin (TTR) in plasma, which prevents its glomerular filtration and its binding to other molecules [2,3]. Currently, there are only two receptors that are known to be RBP4-specific which are as follows: “stimulated by retinoic acid 6” (STRA6) and “stimulated by retinoic acid 6-like” (STRA6L) [4,5,6]. RBP4 signalling through these receptors (JAK/STAT pathway) [7] has been associated with several inflammatory and metabolic diseases [8,9]. Nonetheless, RBP4–mediated pro-inflammatory effects have also been related to the signalling pathways of Toll-like receptors (TLRs), such as TLR4, at least in macrophages [10,11] and cardiomyocytes [12].

High levels of RBP4 have been associated with inflammation in metabolic disorders like insulin resistance [9,13], metabolic syndrome [14,15], fatty liver [16], cardiovascular diseases like atherosclerosis [17], and chronic inflammatory diseases such as psoriasis [8]. Moreover, altered levels of RBP4 have been reported in patients with rheumatic diseases, including rheumatoid arthritis (RA) [18,19] and ankylosing spondylitis [20]. RBP4 was also specifically related to RA co-morbidities in RA patients [21,22]. Furthermore, RBP4 has recently been described to be present within the joints of osteoarthritis (OA) patients, and its levels have been correlated with the production of matrix-degrading enzymes in OA chondrocytes [23]. Nonetheless, RBP4 has not been studied in the context of crystalline arthropathies, which are a group of inflammatory arthritides associated with crystal deposition in the synovial space including gout, which is common in patients with obesity or metabolic alterations [24], and pseudogout (calcium pyrophosphate dihydrate crystal deposition (CPPD)-mediated disease), which is associated with degenerative joint diseases [25,26]. In this study, we aimed to determine the presence of RBP4 in synovial fluid from crystalline arthritis (CA) patients as well as its potential as an articular pro-inflammatory and pro-catabolic factor.

## 2. Materials and Methods

### 2.1. Immunoenzymatic Assay of RBP4

Synovial fluid was obtained through arthrocentesis from twenty-seven people (twenty-one men and six women with an average age of 65.7 ± 14.9 years) who had an acute knee effusion due to CA. All patients provided signed informed consent approved by the Research Ethics Committee of Santiago de Compostela, which is affiliated with the regional public health system (SERGAS) and has a reference number of 2016/258. Samples were treated with type I-testicular bovine hyaluronidase (Sigma-Aldrich, St. Louis, MO, USA) (1500 IU/mL), incubated for 30 min at 37 °C, and centrifuged for 10 min at 410× *g*. Synovial fluid levels of RBP4 were determined employing the human RBP4 ELISA kit (ThermoFisher Scientific, Madrid, Spain) and following the manufacturer’s instructions.

### 2.2. Patients’ Clinical Parameters and Drug Consumption

The clinical parameters of the patients involved in this study of synovial and blood glucose levels, uric acid levels, and leukocyte cell count were obtained from their medical records. Patients’ drug consumption information was also obtained from their medical records. None of the patients were taking any corticosteroids or any medication that could affect uric acid levels at the moment of the extraction. The techniques used by the central hospital laboratory to obtain the clinical parameters were as follows: patients’ synovial and blood glucose levels were obtained through the glucose-hexokinase II method using an ADVIA Chemistry XTP system (Siemens, Munich, Germany); uric acid levels were obtained through the Fossati enzymatic reaction using an ADVIA Chemistry XTP system; and leukocyte cell count was obtained through flow cytometry using an ADVIA 2120i (Siemens).

### 2.3. Molecular Docking Analysis

A molecular docking is used to get a computational approximation of the strength of interaction between two proteins or protein molecules. RBP4 (PDB-3FMZ) and TLR4 with its co-receptor MD2 (PDB-3FXI) structures were obtained from the RCSB Protein Data Bank (PDB) [27] and optimized using PyMol software 2.5.228 [28]. Pydock [29] was used for the molecular docking analysis. Results are shown in Gibbs free energy units (Kcal/mol) and ranked from lowest (strongest interaction) to highest (weakest interaction) for all the possible conformations.

### 2.4. Cell Isolation and Culture

Primary human OA chondrocytes (hOACs) were isolated from cartilage samples from male patients (aged 65–71 years old) who underwent articular-related surgery and provided signed informed consent approved by the Research Ethics Committee of Santiago de Compostela (2016/258). A sample was processed using pronase and collagenase digestion and grown in DMEM/F12 with 10% fetal bovine serum (FBS), 50 μg/mL streptomycin, 4 mM L-glutamine, and 100 U/mL penicillin/streptomycin (P/S) (10%FBS/DMEM/F12) (all from Sigma-Aldrich). All the hOACs used in this study were used under passage three. ATDC5 cells, a chondrogenic cell line, were obtained from the Riken Bank (RCB Cat# RCB0565, RRID:CVCL_3894) and grown in DMEM/F12 with 5% FBS, 4 mM L-glutamine, 100 U/mL P/S, 50 μg/mL streptomycin, 10 µg/mL human apo-transferrin, and 30 nM sodium selenite (Sigma-Aldrich) on adherent culture dishes (Corning, Glendale, AZ, USA). SW982 cells were obtained from ATCC (Cat# HTB-93, RRID:CVCL_1734) and grown in 10%FBS/DMEM/F12. Polymorphonuclear leukocytes (PMNs) were isolated from human blood samples from three healthy donors (one male and two females, 27.3 ± 1.0 years old) who provided signed informed consent approved by the Research Ethics Committee of Santiago de Compostela (2020/357). PMNs where isolated employing Ficoll (ThermoFisher Scientific), dextran (Sigma-Aldrich), and ACK buffers (Lonza, Porriño, Spain) and cultured in RPMI (Sigma-Aldrich) with 10% FBS, 1% P/S, and 4 mM L-glutamine. 

### 2.5. RBP4 Stimulus

ATDC5 and hOAC cells were seeded on 24-well plates (Corning), FBS-deprived for 18 h, and stimulated with 2.5–25 µg/mL mouse RBP4 (Sino Biological, Beijing, China) for 48 h. Once we established an effective dose for RBP4, its synergy with IL1β (Sigma-Aldrich) was tested at the effective concentrations of 25 µg/mL (RBP4) and 1 ng/mL (IL1β). RBP4 inflammatory potential was also tested after 3 h pre-treatment with 1 µM CLI-095 (IBIAN Technologies, Zaragoza, Spain). 

### 2.6. Gene Expression Analysis

Cells were lysed using TRI Reagent (Sigma-Aldrich) and RNA was purified using E.Z.N.A. total RNA kit I (Omega Bio-Tek, Atlanta, GA, USA) according to the manufacturer’s instructions. RNA was treated with DNase I (Cientisol, A Coruña, Spain) and retro-transcribed employing the High-Capacity cDNA Reverse Transcription Kit (Applied Biosystems, New York, NY, USA). Gene expression levels of RBP4; interleukin 6 (IL6); C-C motif chemokine ligand 2 (CCL2); intercellular adhesion molecule 1 (ICAM1); vascular cell adhesion molecule (VCAM); nitric oxide synthase 2 (NOS2); nicotinamide phosphoribosyltransferase (NAMPT); matrix metalloproteinases (MMPs) 1, 3, and 9; chemokine C-X-C motif ligands (CXCL) 1, 2, 8, and 12; and hypoxanthine phosphoribosyltransferase (HPRT) were assessed through a real-time PCR using iTaq Universal SYBR Green Supermix (BioRad, CA, USA) and primers (Sigma-Aldrich) listed in Appendix A. Data analysis was performed using the MxPro software 4.1.0 (Stratagene, CA, USA). Relative quantification was obtained through the 2^−ΔΔCt^ method, with HPRT as the reference gene. Data were normalized against a non-treated, non-stimulated control for each experiment.

### 2.7. Proteomic Analysis

Cellular proteome from hOACs cultured with or without RBP4 (25 µg/mL) for 48 h were studied through micro-liquid chromatography–mass spectrometry (micro-LC–MS/MS) using a hybrid quadrupole TripleTOF 6600 (Sciex, CA, USA). Both proteomes were identified through the qualitative shotgun data-dependent acquisition (DDA) method [30,31], and protein levels were measured through the quantitative sequential window acquisition of all theoretical mass spectra (SWATH) method [31,32]. A 5% false discovery rate (FDR) and a *p*-value ≤ 0.05 were used to filter the dataset. FunRich software 3.1.3 [33] was used to determine proteome enrichment.

### 2.8. Annexin V-FITC Apoptosis Assay

PMNs were seeded in non-adherent plates, treated with 25 µg/mL RBP4 for 6 h, and stained them with annexin V (Sigma-Aldrich) and propidium iodide (ThermoFisher Scientific) according to the manufacturer’s instructions. Data were acquired with a BD FACSAria flow cytometer (BD Biosciences, Madrid, Spain) and analysed using FlowJo™ software 10.8.1 for Windows (BD Biosciences).

### 2.9. Statistical Analysis

Data were presented as the mean ± standard error of the mean (SEM) for three independent experiments. Statistical analyses were performed using GraphPad PRISM 8 software 8.0.2 (La Jolla, CA, USA). Statistically significant differences between experimental conditions were determined through Student’s *t*-test. A difference was considered significant if the *p*-value was ≤ 0.05.

## 3. Results

### 3.1. RBP4 Is Present in the Synovial Fluid of Patients with Crystalline Arthritis

In this study, we first aimed to see if there was any correlation between RBP4 levels in synovial fluid and other clinical parameters of patients with CA obtained from their medical history using the values closest to the date of sample extraction. RBP4 concentration in the synovial fluid of these patients was between 2.44 and 22.62 µg/mL, with a mean concentration of 11.74 ± 5.85 µg/mL (11.53 ± 2.17 µg/mL in gout patients and 11.94 ± 1.30 µg/mL in CPPD patients). We observed that RBP4 levels correlated positively with the total glucose in synovial fluid (Figure 1A) as well as the blood glucose concentration (Figure 1B). However, these correlations were not significant in women (Appendix A). Among these patients, we also found that hypertensive patients had higher levels of synovial RBP4, with a mean of 18.22 ± 1.07 µg/mL vs. 7.96 ± 4.96 µg/mL in non-hypertensive patients (*p* = 0.0003); however, they were not significant in women once again (Appendix A). When splitting these patients according to their specific pathology, this difference was only preserved in gout patients (*p* = 0.0058), who have a higher frequency of metabolic alterations, but not in pseudogout patients (*p* = 0.6291) (Figure 1C). Additionally, we noticed that synovial fluid leukocyte count, a well-known marker of inflammation for this type of arthropathy, correlated negatively with serum uric acid levels (Figure 1D) and also with synovial RBP4 levels (Figure 1E), although no significant correlation was found between uric acid and RBP4 levels (Appendix A).

### 3.2. RBP4 Activates the Inflammatory Response of Chondrocytes

RBP4 has been described as exhibiting inflammatory activity in macrophages and endothelial cells [10,34]. Thus, we tested the inflammatory effect of RBP4 on the chondrogenic cell line ATDC5. We tested a range of RBP4 concentrations (2.5, 5, and 25 µg/mL) similar to the ones that we found in the synovial fluid of CA patients, which were similar to the ones described in OA patients [23]. After 48 h of stimulation with RBP4, we observed that different genes related to the inflammatory response (*IL6*, *CCL2*, *ICAM*, *VCAM*, *NOS2*, *NAMPT*, *MMP1*, *MMP3*, and *MMP9*) were significantly upregulated in a dose-response manner in ATDC5 cells (Figure 2A). After validating these results for hOACs (Figure 2B), we decided to investigate whether RBP4 could potentiate the inflammatory responses elicited by IL1β, a cytokine typically found in different arthritides. As shown in Figure 2C, we found that RBP4 synergically upregulated the IL1β-mediated upregulation of several inflammatory genes, such as *CCL2*, *ICAM* and *NOS2*.

### 3.3. TLR4 Is Partially Involved in the Inflammatory Response to RBP4

TLR4 has been described to participate in RBP4-induced inflammation. Hence, we performed a computational recreation of the interaction of these proteins (Figure 3A–C). After several thousands of iterations (Appendix A), the performed molecular docking revealed that the interaction between TLR4 and RBP4 exhibited a very negative free energy, which suggests that RBP4 may interact with TLR4. To confirm this, we stimulated ATCD5 chondrocytes with RBP4 in the presence or absence of the specific TLR4 inhibitor CLI-095. As shown in Figure 3D, CLI-095 pretreatment inhibits the expression of most of the inflammatory genes induced by RBP4, including *IL6*, *ICAM*, *VCAM*, *NOS2*, *NAMPT*, and *MMP9*. 

### 3.4. RBP4 Expression Is Not Modulated by Inflammatory Stimuli of Chondrocytes

In addition, we studied RBP4 expression under different inflammatory conditions in ATDC5 cells. Results obtained showed that LPS, IL1β, RBP4, and the combination of IL1β and RBP4 did not induce RBP4 mRNA expression. Interestingly, indomethacin, a well-known COX1/2 inhibitor, did not have any effect on RBP4 expression either (Figure 3E). 

### 3.5. RBP4 Induces Proteomic Changes Associated with Increased Glycolysis, Decreased Mitochondrial Metabolism, and Reduced Anabolism in hOACs

Considering that RBP4 induced the expression of inflammatory and catabolic genes in hOACs, we performed a proteomic analysis to achieve further insights into its effects on these cells. After 48 h of treatment with RBP4 (25 µg/mL), 160 proteins were significantly modulated (Appendix A). Among the identified proteins, we found several enzymes involved in two interconnected metabolic pathways of glycolysis and the pentose phosphate pathway (PPP) (Figure 4A). Regarding glycolysis, we observed that fructose bisphosphate aldolase A and C (ALDOA, ALDOC), triosephosphate isomerase (TPI), glyceraldehyde 3-phosphate dehydrogenase (GAPDH), and alpha enolase 1 (ENO1) were significantly augmented in hOACs after RBP4 treatment. Regarding the PPP, 6-phosphogluconate dehydrogenase (PGD) was upregulated, while 6-phosphogluconolactonase (6PGL) was downregulated. There were also two proteins related to these pathways that were significantly increased after RBP4 treatment as follows: glucose 1,6-bisphosphate synthase (PGM2) and aldo-keto reductase (AKR). PGM2, which was increased 18.74 times, is an enzyme that produces more substrate molecules for the glycolytic pathway. In a similar way, isoforms B1, C1, and C2 of AKR, with fold induction changes of 4.36, 2.16, and 1.95, provide more substrate molecules into both glycolysis and the PPP.

In the mitochondrial environment, we could identify two different events. Firstly, many proteins involved in mitochondrial metabolism and oxidative phosphorylation were significantly downregulated, including aspartate aminotransferase (AST), glutamate dehydrogenase 1 (GLUD1), ATP synthase F(0) complex subunit B1 (ATP5PB), and phosphoenolpyruvate carboxykinase (PCK2). On the other hand, two proteins involved in ROS scavenging, mitochondrial superoxide dismutase (SODM) and microsomal glutathione S-transferase (MGST1), were upregulated. Cytosolic enzymes involved in redox regulation, glutaredoxin-1 (GLRX1) and superoxide dismutase (SOD), were also increased. However, heme oxygenase 1 (HMOX1), which is induced by oxidative stress and prevents the activation of catabolic pathways elicited by inflammatory mediators, was 50% downregulated after RBP4 treatment. It is also important to note that upon hOAC stimulation with RBP4, there was also a strong downregulation of multiple structural collagens, like type I, VI, and XII, and an upregulation of catabolic interstitial collagenase (Figure 4C). Moreover, consistent with the inflammatory role that we described for RBP4, there was an upregulation of many interferon-related proteins (Figure 4D). 

Employing the data obtained from the proteomic analysis about the modulated expression of diverse proteins, we performed different types of enrichment analyses. Biological pathway enrichment showed the upregulation of glucose transport and metabolism and the downregulation of insulin receptor signalling and citric acid cycle (Figure 5A). In the cellular component enrichment, we observed that extracellular matrix proteins, including several collagens (types I, III, VI, XII, and XIV), were reduced (Figure 5B). The clinical phenotype enrichment showed that the proteomic profile of the RBP4-treated hOACs exhibited significant similarities with arthritis and diabetes mellitus clinical proteomic profiles (Figure 5C). 

### 3.6. RBP4 Induces the Expression of Inflammatory and Chemoattractant Genes in Synoviocytes

To provide a better context for RBP4’s impact on joints, we also tested its effect on the human synovial cell line SW982. In these cells, although RBP4 downregulated the mRNA expression of *VCAM*, it induced the mRNA expression of other inflammatory and catabolic factors like *IL6*, *ICAM*, *NOS2*, *NAMPT*, *MMP1*, *MMP3*, and *MMP9*. In addition, RBP4 also strongly induced the mRNA expression of key neutrophil chemoattractant genes from the CXCL family, including *CXCL1*, *CXCL2*, and *CXCL8*, while it slightly downregulated the homeostatic chemokine *CXCL12* (Figure 6A).

### 3.7. RBP4 Promotes Neutrophils Activation

Taking aim at the DAMP-like activity of RBP4 and its induction of neutrophil chemoattractant genes in synoviocytes, we studied the effect of RBP4 on the vitality of primary human PMNs by annexin V-FITC apoptosis assay. After 6 h stimulation with RBP4, PMN viability was significantly increased in comparison with the untreated cells, whereas the number of cells in early and late apoptosis was reduced (Figure 6B,C).

## 4. Discussion

In this study, we described the concentration of RBP4 in the synovial fluid of patients with CA, its positive correlation with glucose levels, and its negative correlation with synovial leukocyte count. In vitro, RBP4 activated neutrophils and exhibited inflammatory and chemotactic activity in hOACs and synoviocytes through TLR4 activation. Moreover, it induced proteomic changes in hOACs consistent with an inflammatory response, increased catabolism, and decreased anabolism, mitochondrial dysfunction, and significant glycolysis promotion. This supports the role of RBP4 as a DAMP not only in CA but also in other diseases like OA [35]. 

Rheumatic diseases are closely related to metabolic alterations like diabetes, non-alcoholic fatty liver, obesity, and metabolic syndrome [36,37]. RBP4 is an hepatokine that has been related to these diseases [9,14,15,16] and also to other pathologies related to inflammation [8,38] and articular tissue destruction [18,19,20], including OA [23]. RBP4 has inflammatory properties across diverse cell types [10,39,40] and activates TLR4-mediated responses in adipocytes [10]. Despite TLR4 being one of the most important receptors of the innate immunity and having a key role in the development of several rheumatic diseases [41], there are no studies on the specific effect and signalling of RBP4 on different articular cells like chondrocytes or synoviocytes. Moreover, although the concentration of RBP4 in OA synovial fluid has been previously described, little was known about its concentration in CA synovial fluid. It is well known that crystalline arthritides have a bigger inflammatory component than OA. Nonetheless, the concentration range in these patients was between 2.44 and 22.62 µg/mL, which overlaps with the concentration range described in OA patients [23], thereby suggesting that the articular inflammatory environment does not determine RBP4 synovial levels. In line with this, RBP4 levels were also not significantly different between pseudogout and gout patients. Consistent with this idea, neither inflammatory stimuli nor anti-inflammatory drugs were able to modulate the RBP4 mRNA expression in chondrocytes in vitro. Even though the levels of synovial RBP4 in healthy controls remain to be determined, its median plasma level has been determined in a cohort of 150 healthy individuals [42]. Given that plasma RBP4 levels has been described to be almost 2.5 times higher than synovial levels [23], it is possible to speculate that synovial RBP4 levels in healthy controls (15.5 µg/mL) in plasma [42] might be lower than those in CA and OA patients.

RBP4 is tightly related to several rheumatic disease comorbidities, like diabetes and cardiovascular pathologies [43]. Therefore, we investigated whether the metabolic context of CA patients could be related to the synovial concentration of RBP4. We found a significant correlation between synovial RBP4 levels alongside total synovial glucose and the blood glucose concentration. This finding is consistent with previous reports, where RBP4 contributed to insulin resistance and glucose intolerance at the local and systemic levels [13,39,44]. We also observed that CA patients with hypertension exhibited higher synovial RBP4 levels than normotensive patients. Supporting this, RBP4 systemic levels were described to correlate with the increased prevalence of hypertension [43]. Interestingly, this significant increase in RBP4 levels in CA patients with hypertension was observed in gout patients but not in pseudogout patients, which might be explained by the fact that gout is associated with more metabolic alterations than pseudogout [25]. 

Metabolic alterations drive the worsening of multiple rheumatic diseases. Hence, the increased synovial levels in CA patients with elevated glucose levels and hypertension led us to investigate its effects on articular cells in vitro. Consistent with the inflammatory effect of RBP4 in extraarticular tissues, the RBP4 concentrations observed in the synovial fluid of CA patients promoted the expression of multiple inflammatory and catabolic genes in cultured hOACs and synoviocytes. Interestingly, RBP4 synergized with IL1β in chondrocytes, thereby promoting the expression of several inflammatory genes (*CCL2*, *ICAM*, and *NOS2*). This has not been described before but may be in concordance with other studies that showed IL1β upregulation in adipocytes upon RBP4 stimulation [10]. Overall, the strong inflammatory effect of RBP4 and its synergic activity with IL1β might explain the contribution of certain metabolic diseases to the development of rheumatic diseases.

To achieve further insights into the mechanism of RBP4 inflammatory actions, we explored RBP4 potential binding to the TLR4 receptor through computational chemistry. Data obtained revealed that RBP4 directly docks to the TLR4 receptor. This suggests a direct activation of this receptor by RBP4, which supports previous findings about RBP4 signalling [10]. Apart from the receptor itself, TLR4 agonist recognition can vary depending on the cell type [45]. In chondrocytes, TLR4-specific inhibition with CLI-095 partially inhibited the inflammatory actions of this hepatokine, which points to RBP4 as an articular DAMP.

Surprisingly, despite the fact that RBP4 promoted cellular inflammatory responses, we found that RBP4 levels correlated negatively with leukocyte count in the synovial fluid of CA patients. However, even though synovial leukocyte count is an inflammatory sign, all of these patients had a cell count of over 5000, which is considered a marker of an articular inflammatory process [46]. Moreover, a reduced leukocyte count has been associated with the phagocytosis of urate crystals [47]. In line with this, we found that uric acid levels also correlated negatively with leukocyte count. Furthermore, since the synovial tissue controls the number of infiltrating cells entering into the articular compartment, we explored the effect of RBP4 on human synoviocytes in vitro. RBP4 stimulation of these cells upregulated the expression of the neutrophil chemoattractant genes, while the homeostatic chemokine *CXCL12* was significantly downregulated. This might produce leukocyte retention in the synovial membrane [48], which would explain the negative correlation between RBP4 and the synovial fluid leukocyte count observed in CA patients. All these data together led us to hypothesise that part of these cells could still be in the joint, but in the synovial membrane rather than in the synovial fluid. Nonetheless, to discard a pro-apoptotic effect of RBP4 on the infiltrating cells, we tested the effect of RBP4 on cultured human primary neutrophils, which are the main infiltrating cells in CA joints. We observed that RBP4 stimulation of these cells promoted their activation, increasing their vitality and reducing their apoptosis, which further supports the role of this protein as a DAMP [49].

Cartilage degradation is a hallmark of the progression of multiple rheumatic diseases because it powerfully impairs articular function. Apart from RBP4 inflammatory actions in hOACs, we thus explored its effect on their metabolism upon in vitro stimulation. We found that RBBP4 induced proteomic changes associated with the promotion of the glycolytic pathway. This leads to an accumulation of pyruvate that has been described to be related to inflammatory response [50,51,52] and also to cartilage degradation, as has been reported in OA cartilage [53]. Moreover, proteomic results also indicated the development of a relevant mitochondrial dysfunction upon hOAC stimulation with RBP4. The mitochondrial alterations included the upregulation of ROS-related enzymes along with the downregulation of oxidative phosphorylation enzymes. Chondrocyte mitochondrial dysfunction is widely known to be a key step toward cartilage degradation [54]. Consistent with RBP4 activation of TLR4, a decrease in mitochondrial activity together with an increase of ROS generation have been described as a result of the activation of the innate immune responses [55,56]. 

Altogether, the metabolic and mitochondrial alterations observed in RBP4-stimulated hOACs suggest the presence of a pro-catabolic environment. In line with this, proteomic data as well as diverse enrichment analysis confirmed a strong inhibition of the expression of multiple protein matrix components. Moreover, a pathological enrichment analysis revealed that the proteomic profile elicited by RBP4 in hOACs was similar to those associated with diabetes and arthritis, which further supports that RBP4 might be a common link between systemic metabolic imbalance and articular inflammatory and catabolic events. Of note, synovial RBP4 levels have been observed to correlate with its systemic levels in OA patients [23]. Supporting the key role of RBP4 in metabolic diseases, its therapeutic targeting has successfully controlled the development of hypertension and insulin resistance in animal models [43]. Accordingly, considering the deleterious effects of RBP4 in diverse articular cells, we hypothesise that RBP4 would be an appealing therapeutic target to control the impact of diverse RBP4-associated metabolic comorbidities in the joint destruction associated with CA and other arthritides.

Despite all the data shown, we acknowledge that this study has some limitations. First, this study does not have an animal model that could have underpinned our results. However, we are confident that our results with human primary cells provide enough evidence to situate RBP4 as a DAMP. Finally, it is important to note that the difference between male and female sample numbers could be a limitation. Nonetheless, this imbalance is consistent with the grater prevalence of these diseases in males [57].

In conclusion, in this work we confirmed the presence of relevant concentrations of RBP4 in the synovial fluid of CA patients that are related to the metabolic and cardiovascular contexts of these patients. We provide evidence that RBP4 works as DAMP at these concentrations, inducing strong inflammatory and metabolic responses in chondrocytes, synoviocytes, and neutrophils. These effects might explain the contribution of certain metabolic diseases to joint destruction in diverse rheumatic diseases like crystalline arthritis.

## Figures and Tables

**Figure 1 antioxidants-13-00124-f001:**
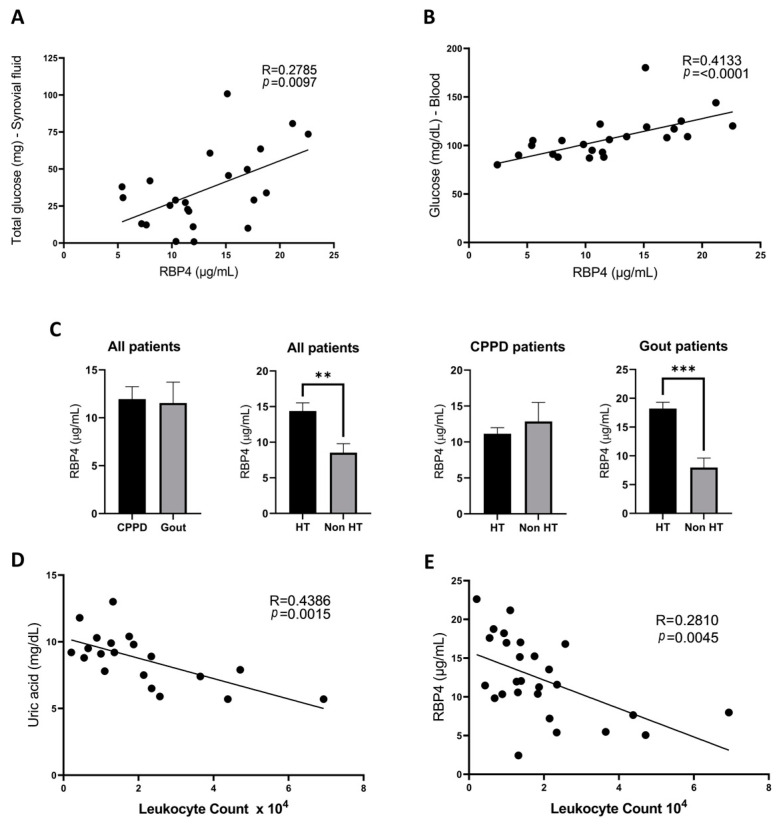
Correlations found in RBP4 levels tested through an ELISA assay used for the analysis of synovial fluid from 27 crystalline arthritis patients (14 patients with gout and 13 patients with CPPD). There is a positive correlation between patients’ synovial RBP4 concentrations and total glucose in the synovial fluid (**A**) and blood glucose levels (**B**). There was no significant difference in synovial RBP4 levels in gout and CPPD patients (*p* = 0.88). However, when separating our patients based on hypertension diagnosis, we observed that patients’ RBP4 levels are higher in hypertensive patients when both considering all patients together and gout patients alone, but this difference is absent in CPPD patients (**C**). Patients RBP4 levels correlated negatively with the synovial leukocyte count (**D**), which also correlated negatively with uric acid blood levels (**E**). RBP4 (retinol binding protein 4), CPPD (calcium pyrophosphate deposition). (*p* = statistical significance; ** *p* < 0.01; *** *p* < 0.001).

**Figure 2 antioxidants-13-00124-f002:**
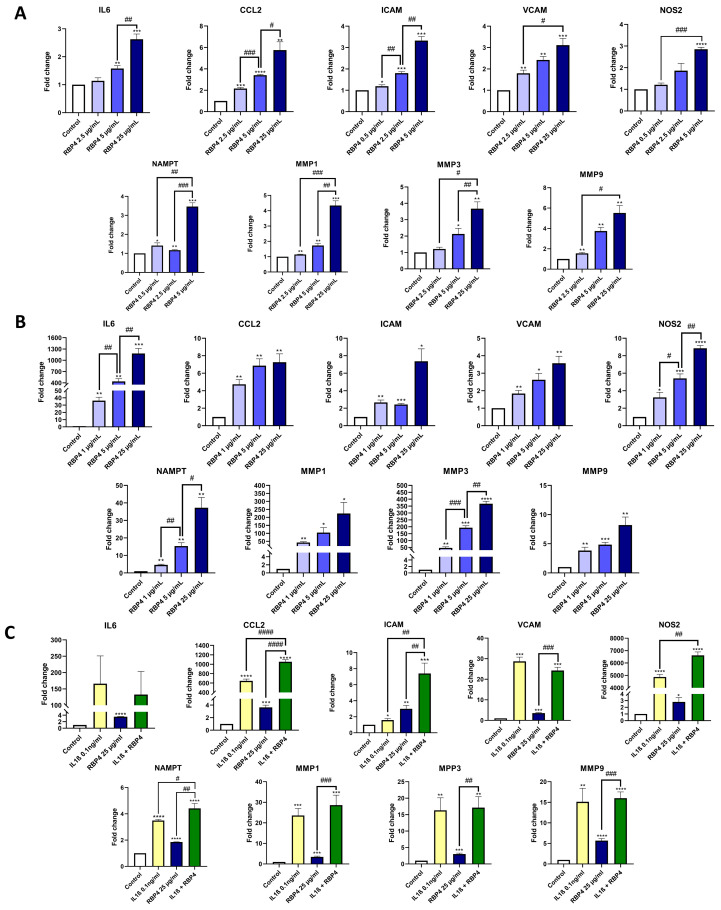
RBP4 stimulation of ATDC5 cells for 48 h produced an upregulation of *IL6*, *CCL2*, *ICAM*, *VCAM*, *NOS2*, *NAMPT*, *MMP1*, *MMP3*, and *MMP9* in a dose-response way (**A**). Human OA primary chondrocytes were treated with RBP4 to validate the previous results, obtaining the same upregulation of all the inflammatory genes in a dose-response way (**B**). RBP4 can synergize with IL1β, upregulating *CCL2*, *ICAM*, and *NOS2* independently and also synergistically. *NAMPT* and *MMP1* showed an additive response. *VCAM*, *IL6*, *MMP9*, and *MMP3* were upregulated by both stimuli, but they had no synergic or additive effect together (**C**). RBP4 (retinol binding protein 4), IL6 (interleukin 6), CCL2 (C-C motif chemokine ligand 2), ICAM (intercellular adhesion molecule 1), VCAM (vascular cell adhesion molecule), NOS2 (nitric oxide synthase 2), NAMPT (nicotinamide phosphoribosyltransferase), MMP1 (matrix metallopeptidase 1), MMP3 (matrix metallopeptidase 3), and MMP9 (matrix metallopeptidase 9), IL1β (Interleukin 1 Beta). (* *p* < 0.05; ** *p* < 0.01; *** *p* < 0.001; **** *p* < 0.0001 versus control condition. # *p* < 0.05; ## *p* < 0.01, ### *p* < 0.001; #### *p* < 0.0001 between the conditions marked).

**Figure 3 antioxidants-13-00124-f003:**
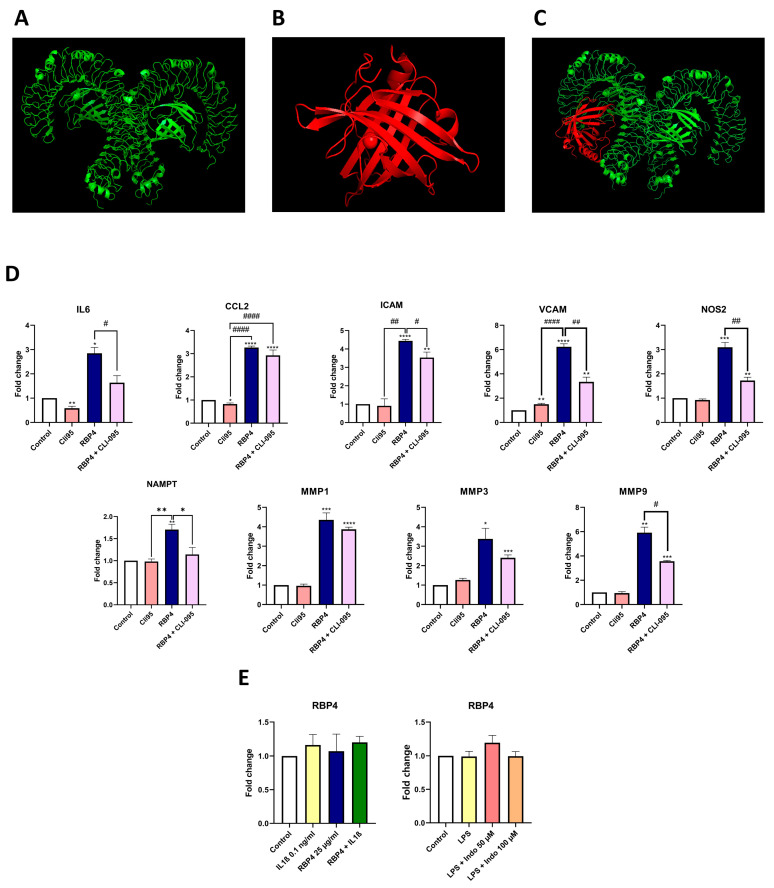
Visual representation of TLR4 structure in green (**A**), RBP4 in red (**B**), and the best result of the molecular docking (**C**). Cli95 pre-treatment partially inhibits the effect of RBP4, downregulating *IL6*, *ICAM*, *VCAM*, *NOS2*, *NAMPT*, and *MMP9* (**D**). RBP4 expression is not modulated by IL1β, RBP4, or both stimuli together. The inflammatory stimulus LPS alone or in combination with a well-known COX1-2 inhibitor (indomethacin) in different doses cannot modulate RBP4 expression (**E**). TLR4 (Toll-like receptor 4), RBP4 (retinol binding protein 4), IL6 (interleukin 6), CCL2 (C-C motif chemokine ligand 2), ICAM (intercellular adhesion molecule 1), VCAM (vascular cell adhesion molecule), NOS2 (nitric oxide synthase 2), NAMPT (nicotinamide phosphoribosyltransferase), MMP1 (matrix metallopeptidase 1), MMP3 (matrix metallopeptidase 3), and MMP9 (matrix metallopeptidase 9), IL1β (Interleukin 1 Beta), Indo (indomethacin). (* *p* < 0.05; ** *p* < 0.01; *** *p* < 0.001; **** *p* < 0.0001 versus control condition. # *p* < 0.05; ## *p* < 0.01; #### *p* < 0.0001 between the conditions marked).

**Figure 4 antioxidants-13-00124-f004:**
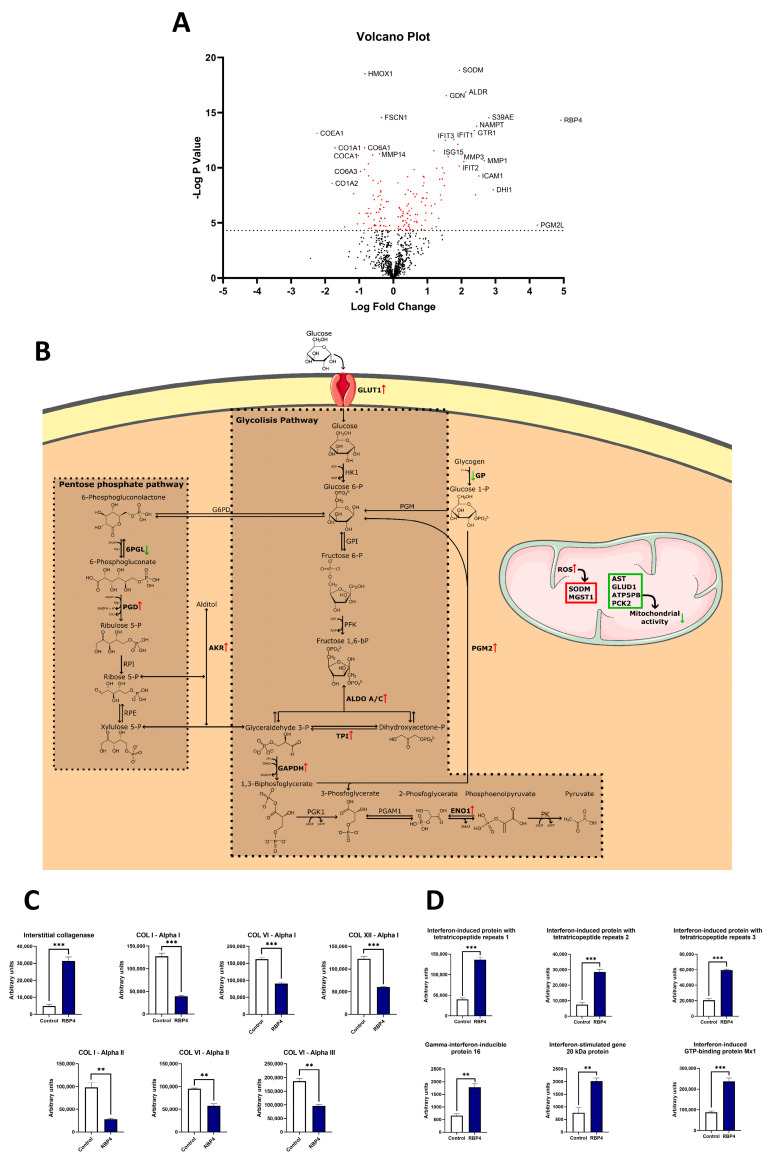
(**A**) Volcano plot showing a general view of the proteomic analysis, red dots represent the proteins that are significantly regulated by the RBP4 treatment while black dots are the proteins that are not significantly different. (**B**) RBP4 stimulation enhances glucose uptake through the GLUT1 receptor and produces a general upregulation of the glycolytic pathways, including ALDO A, ALDO C, TPI, GAPDH, and ENO1, which were enriched in RBP4-treated cells. In the pentose phosphate pathway, there was an enrichment of 6PGL and a depletion of PGD. Two proteins highly related to the previous pathways are PGM2 and AKR1. PGM2 generates more glucose 6-P in the glycolytic process, and AKR1 can produce more aldoses or more alcohol-enriched sugars that might increase the oxidative stress in the cell. (**C**) Graphic representation of the different collagen-related proteins that are significantly changed between control and RBP4-treated chondrocytes in the proteomic analysis. (**D**) All the interferon-related proteins are highly upregulated. RBP4 (retinol binding protein 4), GLUT1 (glucose transporter 1), ALDO A (fructose-bisphosphate aldolase A), ALDO C (fructose-bisphosphate aldolase C), TPI (triosephosphate isomerase), GAPDH (glyceraldehyde 3-phosphate dehydrogenase), 6-phosphogluconolactonase (6PGL), ENO1 (enolase 1), PGD (6-phosphogluconate dehydrogenase), PGM2 (glucose 1, 6-bisphosphate synthase), AKR (aldo-keto reductase). (Green arrows represent downregulated proteins; Red arrows represent upregulated proteins. ** *p* < 0.01; *** *p* < 0.001).

**Figure 5 antioxidants-13-00124-f005:**
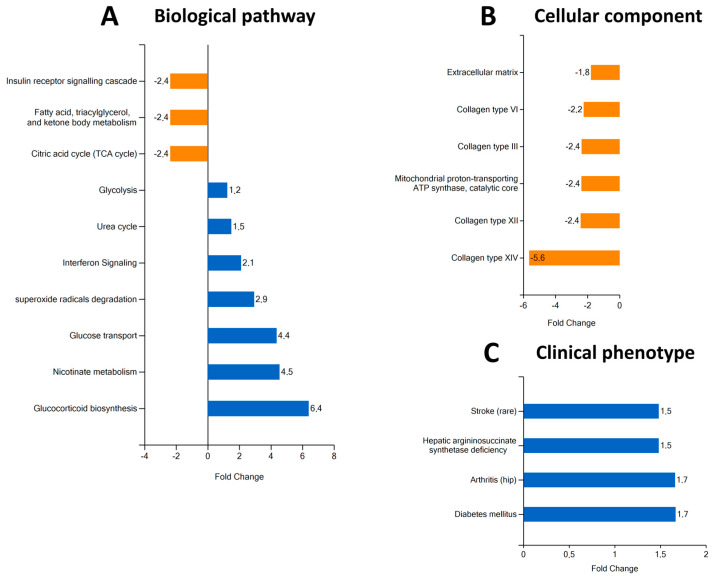
Different types of proteomic enrichments after RBP4 treatment. Pathway enrichment analysis (**A**) shows that there is a depletion of insulin receptor signalling cascade, fatty acid metabolism, and citric acid cycle pathways. It also shows an enrichment of glycolysis, urea cycle, interferon signalling, superoxide radical degradation, glucose transport, nicotinate metabolism, and glucocorticoid biosynthesis pathways. Cellular component analysis (**B**) shows a general depletion of collagens (types I, III, VI, XII, and XIV), extracellular matrix, and mitochondrial proton-transporting ATP synthase complexes. Clinical phenotype analysis (**C**) shows similarities between the proteomic profile of RBP4-stimulated chondrocytes and hepatic argininosuccinate synthase deficiency, arthritis, and diabetes mellitus. RBP4 (retinol binding protein 4).

**Figure 6 antioxidants-13-00124-f006:**
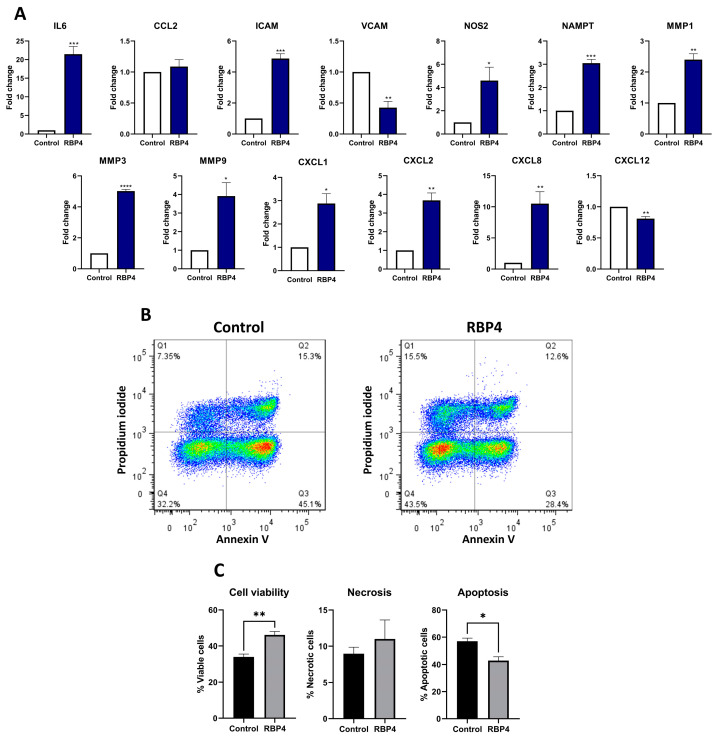
Inflammatory genes expression on a human synovial cell line (SW982) after RBP4 treatment. *IL6*, *ICAM*, *NOS2*, *NAMPT*, *MMP1*, *MMP3*, and *MMP9* were upregulated after RBP4 treatment, but VCAM was downregulated, while *CCL2* showed no changes. Cellular adhesion-related genes *CXLC1*, *CXCL2*, and *CXCL8* were upregulated, while *CXCL12* was downregulated (**A**). A representative image of the flow cytometry experiment. RBP4 produced an activation of neutrophils and reduced apopotosis and the canonical death pathway of neutrophils, but it did not significantly increase the necrotic pathway. Each dot represents one cell and the colour scale goes form red (more density of cells) to blue (less density of cells) (**B**). Graphic representation of the neutrophil activation by RBP4 experiment in figure **B** (**C**). RBP4 (retinol binding protein 4), IL6 (interleukin 6), CCL2 (C-C motif chemokine ligand 2), ICAM (intercellular adhesion molecule 1), VCAM (vascular cell adhesion molecule), NOS2 (nitric oxide synthase 2), NAMPT (nicotinamide phosphoribosyltransferase), MMP1 (matrix metallopeptidase 1), MMP3 (matrix metallopeptidase 3), MMP9 (matrix metallopeptidase 9), CXCL1 (chemokine (C-X-C) motif) ligand 1), CXCL2 (chemokine (C-X-C motif) ligand 2), CXCL8 (chemokine (C-X-C motif) ligand 8), and CXCL12 (chemokine (C-X-C motif) ligand 12). (* *p* < 0.05; ** *p* < 0.01; *** *p* < 0.001; **** *p* < 0.0001).

## Data Availability

Data is contained within the article and Appendix A.

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
