# Peer review of "The Hepatokine RBP4 Links Metabolic Diseases to Articular Inflammation"

_antioxidants, 2024, doi:10.3390/antiox13010124_

Round 1
Reviewer 1 Report
Comments and Suggestions for Authors
This manuscript reports pro-inflammatory and catabolic roles, as well as altered metabolic pathways induced by RBP4, an hepatokine, in human osteoarthritic chondrocytes, chondrocytic cell lines and a synovial cell line. Moreover, the authors report the existence of a positive correlation between synovial fluid levels of RBP4 and synovial fluid and blood glucose levels in patients with crystalline arthritis. RBP4 levels were found to be higher in hypertensive than in non-hypertensive patients, namely in those with gout, while no differences were found in patients with Calcium pyrophosphate deposition arthritis. Using a variety of techniques, the authors found that RBP4 likely acts as a DAMP through activation of TLR4 and conclude that RBP4 may play a relevant role in linking metabolic, cardiovascular and arthritic diseases. The study is well planned and executed and only a few points need to be elucidated to make conclusions more robust, as follows:
1) Isn't the Research Ethics Committee from Santiago de Compostela affiliated to any academic or clinical institution? Please, provide details about the identity of the Ethics Committee. Also, the date of approval seems to be 2016. Is the approval still valid?
2) Were primary chondrocytes passaged? At what passages were the cells used?
3) Was the hOAC donnor hypertensive, diabetic or did he have any other metabolic imbalance?
4) Since the manuscript is mainly focused on cristalline arthritis, it would be important to further strengh the correlation between RBP4 and those conditions to assess the correlation between RBP4 and uric acid blood levels.
5) Why was MMP3 chosen to assess catabolic efffects, instead of MMP-13 which is much strongly related to OA?
6) Page 6, line 221: I think that "figure 3D" should be figure 3E.
7) Downregulation of coll I may signify less chondrocyte dedifferentiation or progression to hypertrophy, since coll I is not an articular cartilage native collagen. Were no differences detected in coll II levels? On the other hand since collagens are secreted, can the decreased levels of intracellular coll I be due to increased secretion?
8) OA is more prevalent in post-menopausal women and estrogens have significant effects in metabolism. It would be very interesting to test the effects of RBP4 in female chondrocytes in the presence and absence of estrogens. It would also be very interesting to assess potential correlations between synovial fluid RBP4 levels, synovial fluid and blood glucose and hypertension in men and women separately.
Comments on the Quality of English Language
Reviewer 2 Report
Comments and Suggestions for Authors
This is a very interesting finding, with many data included in one manuscript. However, the presentation should be improved as the structure of the manuscript should be clarified (with in vivo results and effects of RBP4 on chondrocytes).
Concerning the in vivo results, especially the difference between CPPD and gout patients should be better presented (with number of patients and assays performed especially in figure 1c).
Concerning the effects of RBP4 on chondrocytes, figures should be larger and legends more detailed.
Major comments:
The abstract should include the concept of in vivo and in vitro results including significant data.
Line 84: At least 3 assays with isolation of primary human OA chondrocytes should be performed.
Patients' characteristics should be outlined including the medication (concerning uric acid lowering and corticosteroids) and other comorbidities observed. What are the findings in normal, healthy controls to compare with?
The discussion should clarified according to the in vivo / in vitro concept. The limitations of this study should be outlined.
The conclusion should be more detailed, based on the results from above.
Minor comments:
Line 14, "in an articular context": please specify using the crystalline arthritis (CA).
Line 26 "linking them to the patients' metabolic and cardiovascular context": please focus on conclusions definitely drawn from the results.
Line 28: please specify which articular cells are meant.
Line 76, heading: Use "Molecular docking analysis" instead of "Docking". In the paragraph, the procedure should be shortly introduced, e.g. "Docking is used to predict ..."
The cell lines should be characterised in the methods section: e.g. chondrogenic cell line ATDC5 etc.
Line 103: Which cells were seeded?
Lines 153-157 should be part of the methods section.
Line 194: MMMP9 is not correct.
Line 198: Omit "synergic".
Line 205 and the whole manuscript: check the references for the correct form.
Line 323: Legend of figure 6 should be improved with more details to explain the figure.
Line 352: Please specify "articular" cell again!
Comments on the Quality of English Language
Check for English language! (line 329 et al.)
Round 2
Reviewer 1 Report
Comments and Suggestions for Authors
I think that the information regarding affiliation of the research committee to the regional health system is relevant and should be included because readers will likely be unfamiliar with the research and health system of Galicia.
The graphs presented in the response letter could be included in the manuscript what would help the reader retain more easily the significant and non-significant correlations. The considerations about gender differences might also be included, along with the limitation due to the small number of female samples. I think that the observations are pertinent, even if not statistically significant because they raise awareness of the importance of gender on biological and clinical responses.
Comments on the Quality of English LanguageOnly minor corrections, like hypertensive instead of "hypertense", are required.
Author Response
I think that the information regarding affiliation of the research committee to the regional health system is relevant and should be included because readers will likely be unfamiliar with the research and health system of Galicia.
Thank you for your suggestion, we have added the information about the regional health system in the manuscript (lines 70-71)
The graphs presented in the response letter could be included in the manuscript what would help the reader retain more easily the significant and non-significant correlations. The considerations about gender differences might also be included, along with the limitation due to the small number of female samples. I think that the observations are pertinent, even if not statistically significant because they raise awareness of the importance of gender on biological and clinical responses.
Thank you for your appreciation, we have added both figures to the supplementary data as “Supplementary Figure 1 and 2” (lines 180, 183 and 190). Also, we have added a section in the discussion about the study limitations where we discuss about the gender disparity in number of samples. As we state in the discussion, this difference in gender number of samples is because it has been described that there are significantly more men affected by gout than women (Evans, P.L et al. Advances in Rheumatology. 2019. PMID: 31234907)
Only minor corrections, like hypertensive instead of "hypertense", are required.
Thank you for the detailed revision of our article. We have revised the manuscript through a C2 certified scientific revisor and through a grammatic AI based software to find all the human-based mistakes and correct them.

Reviewer 2 Report
Comments and Suggestions for Authors
Thank you for resubmission. This is indeed an important issue. However, some points are still unclear and sufficiently answered.
Major comments:
I understand the need to mention CA instead of "articular context" in the objectives, but in the results the number of tested patients with gout and CPPD has to be clarified and the most important data (with mean +/- SEM and p-values) provided to underline the driven conclusions.
Line 84: This point is not really answered by the authors. What is "patients clinical data obtention"? The information on medication has to be added into this paragraph, not to the immunoenzymatic assay! What about consent and vote of the ethics committee?
The number of gout patients in comparison with CPPD patients is still not provided.
Missing healthy data is a typical limitation of the study, which can well be argued in the discussion. The authors refer to a paper with RBP4 serum levels in healthy patients (Li et al. Bioscience Reports. 2018. PMID: 30135138). This information should be provided in detail and also included into the manuscript and discussed?
The discussion on RBP4 data from in vivo and ex vivo are still not really separated, stressing especially the difference between gout and CPPD.
Ethical considerations have to be added.
Comments on the Quality of English LanguageSome mistakes, language editing recommended
Author Response
Thank you for resubmission. This is indeed an important issue. However, some points are still unclear and sufficiently answered.
Major comments:
I understand the need to mention CA instead of "articular context" in the objectives, but in the results the number of tested patients with gout and CPPD has to be clarified and the most important data (with mean +/- SEM and p-values) provided to underline the driven conclusions.
Thank you for your comment. According to your suggestion, we have added the number of patients with gout and CPPD to the legend of Figure 1. We have also added the mean ± SEM of synovial RBP4 levels in gout, CPPD, hypertensive and non-hypertensive patients to results section, as well as the p-values for these comparisons (lines 177-183).
Line 84: This point is not really answered by the authors. What is "patients clinical data obtention"? The information on medication has to be added into this paragraph, not to the immunoenzymatic assay! What about consent and vote of the ethics committee?
Thank you for your comment. In order to clarify this section, we have renamed it. Also, we added more information about the clinical data of patients clarifying that we did not perform the analysis and just took the data from the clinical records (lines 77-88). The information regarding the medication has also been moved to this section. Finally, the information regarding the ethics committee approval is in the first section of the methodology (lines 69-71).
The number of gout patients in comparison with CPPD patients is still not provided.
Thank you for your comment. Following your suggestion, we have added the number of patients with gout and CPPD to the legend of Figure 1.
Missing healthy data is a typical limitation of the study, which can well be argued in the discussion. The authors refer to a paper with RBP4 serum levels in healthy patients (Li et al. Bioscience Reports. 2018. PMID: 30135138). This information should be provided in detail and also included into the manuscript and discussed?
Thank you for your suggestion. We have incorporated the information in the discussion section to explicitly address the limitations of our research. There, we provide in detail the RBP4 blood levels in healthy patients and discuss them in the CA context This addition hopefully will provide a more comprehensive understanding of the constraints and potential areas for improvement in our study (lines 401-406).
The discussion on RBP4 data from in vivo and ex vivo are still not really separated, stressing especially the difference between gout and CPPD.
Thank you for your valuable feedback on our manuscript. The results obtained from the patients are discussed in the first paragraphs of the discussion section. Then, in vitro data is introduced with the sentence “Hence, the increased synovial levels in CA patients with elevated glucose levels and hypertension led us to investigate its effects on articular cells in vitro”. Nevertheless, we have made additional changes to clearly identify the results from in vitro experiments (e.g. “on cultured cells”, “in vitro”, “upon in vitro stimulation”) as opposed to patients’ data (e.g. “in the synovial fluid of CA patients”, “observed in CA patients”). Please see the discussion section in the manuscript.
Regarding the difference between gout and pseudogout (CPPD), we have added this to the discussion section (lines 384-385).
Ethical considerations have to be added.
Thank you for your valuable feedback. The information regarding the ethics committee approval is in the first section of the methodology (lines 69-71). In addition, there is a section of conflict of interests where we declare that we have no conflicts of interest (lines 506-507).
Comments on the Quality of English Language
Some mistakes, language editing recommended
Thank you for the detailed revision of our article. We have revised the manuscript through a C2 certified scientific revisor and through a grammatic AI based software to find all the human-based mistakes and correct them.
Round 3
Reviewer 2 Report
Comments and Suggestions for Authors
-